



# Study on Mechanical Properties and Dissipation Capacity of Ring Net in Passive Rockfall Barriers

Chengqing Liu[1,2], Shuai Tian[3], Chengjie Xu[1], Jingjin Yang[1]

[1] School of Civil Engineering, Southwest Jiaotong University, Chengdu 610031, China.
[2] Key Laboratory of High-speed Railway Engineering Ministry of Education, Chengdu 610031, China.
[3] China Design Group Co., LTD., Nanjing, 210014, China.

*Correspondence to*: Chengqing Liu (lcqjd@swjtu.edu.cn)

**Abstract.** Passive rockfall barriers are used to mitigate the danger of falling rock and debris on steep slopes. The barriers consist of different flexible components including steel beams, cables, braking elements and wire nets. The engineering design of the net is especially important because it is one of the primary energy consuming components of the system. Nets consisting of interwoven wire rings are widespread but can fail in practice. In this paper we investigate the energy dissipation capacity of a single ring under application of two point, four point and six point tensile loading. The correctness of the theoretical results is verified by experiments. A numerical model of the ring net for application in modelling flexible barriers is then presented. We examine different support boundary conditions and rockfall impact angles on barrier response. We find the release of boundary conditions can increase the overall flexibility of the ring net. As the impact angle increases, the impact energy of the rock on the ring net will experience a gradual decline. The derived energy dissipation formulas provide a theoretical basis for engineering rockfall barriers.

## 1 INTRODUCTION

With the changes of climate in recent years, geological disasters occur frequently in the southwest and southeast of China, and the collapse and rockfall occur suddenly, it poses a serious threat to the main traffic lines and towns in mountainous areas (see Fig. 1). The prevention measures of rockfall and soil collapse are mainly divided into active rockfall barriers, passive rockfall barriers, block stone walls, etc. It can't be denied that the passive rockfall barriers had been widely used in practice because of their convenience, small interference, low cost of construction and the protection for environment (see Fig. 2). The passive rockfall barriers, besides, provide flexible safety protection consisting of four parts: a metal flexible net, a fixed system (anchor rod, anchor rope, base and support ropes, etc.), brake rings and steel columns. This structure can reduce or prevent the geological disasters effectively (see Liu et al. 2017).




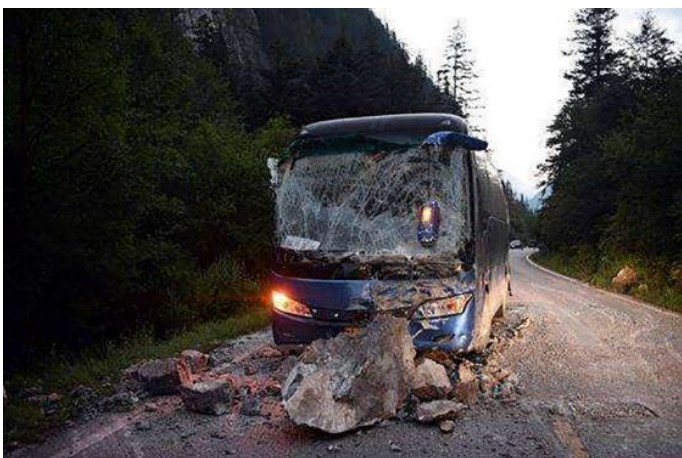

**Fig. 1 Rockfall endangers life and safety**

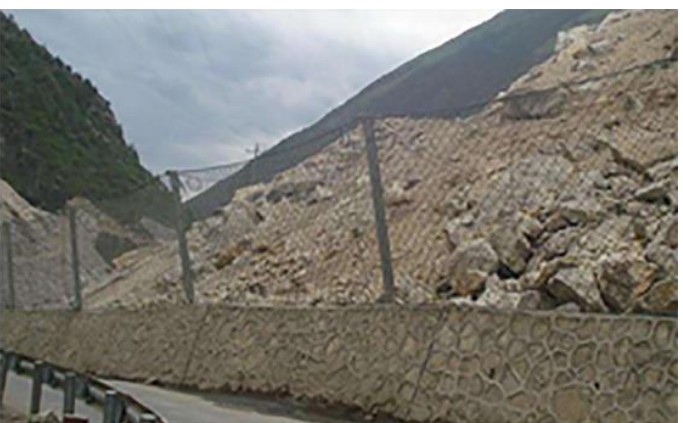

**Fig. 2 Application of Passive Rockfall Barriers in Practical Engineering**

At present, as the main energy dissipation components in the passive rockfall barriers, the widely-used metal flexible nets are usually presented in the following two forms: rhombic net and ring net (see Fig. 3). The ring net has more extensive application than rhombic net because of its flexibility, energy dissipating ability and interconnection of independent ring. However, the ring nets are often destructed in practical engineering. Possible reasons for the destruction are the uncertainty

of collapse and the complicated structural behaviour of the ring net, the basic principle research on ring net is not sufficient. Therefore, the systematic study of the mechanical properties and energy dissipation of ring net are of great importance to the design of metal flexible net.





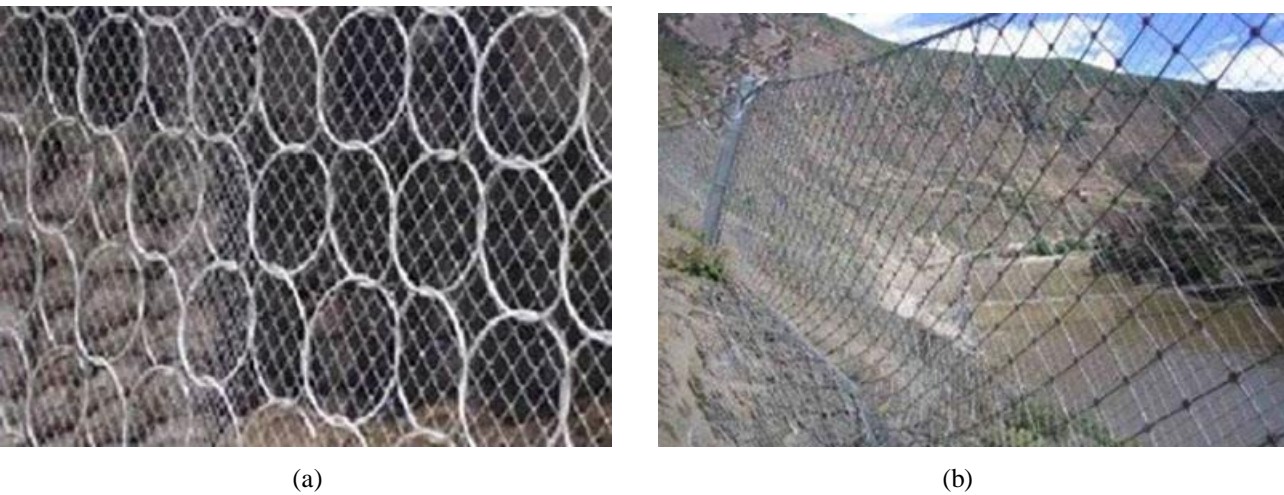

(a)                                                                                      (b)

**Fig. 3  Forms of passive rockfall barriers. (a) Ring net. (b) Rhombic net**

Currently, it is true that the foreign researchers have made great achievements in the structure of passive rockfall barriers. Although there is barely no difference between the domestic and international test study, the research on mechanics is still inadequate. Spadari et al. (2012) proposed a new method to quantitatively characterizing the ''bullet effect''. This approach was based on three strongly correlated dimensionless parameters pertaining to the performance of the mesh (E*), to the ratio of stiffness over strength (S*) and to the geometry of the mesh (G*). Meanwhile, the approach was verified using a numerical model. Thoeni et al. (2013) proposed the discrete contact model of wire rope net with distortion, which can study the mechanical properties of wire rope net in the rockfall barriers more precisely. Hambleton et al. (2013) took the "bullet effect" and the bending deformation of the wire rope net into consideration, after that, a numerical analysis model to determine the critical kinetic energy of the passive rockfall barriers is proposed. Buzzi et al. (2015) studied the effect of block size and meshed geometry (aperture and wire diameter) on the TECCO net performance through experiment, and captured the bullet effect. It was found that decreasing the mesh aperture by 19% improves the performance by 50% while only an extra 30% could be gained by increasing the wire diameter by 33%, so aperture is more important than wire diameter to increase mesh resistance. At the same time, they redefine a dimensionless geometrical parameter G*(the formulation of G* that was proposed in Spadari et al. (2012)) and to validate a simple power type equation relating the mesh characteristics and the mesh performance. Coulibaly et al. (2017) presented an innovative discrete model of steel rings for application in flexible rockfall barriers. The 2-point traction and 4-point traction loading configurations are used as references, an analytical response of the ring model under these loading configurations is derived and a multi-criteria calibration method to obtain the optimal model parameters is developed based on the Levenberg-Marquardt algorithm. Escallon et al. (2014,2015) carried out macroscopic finite element analysis on the modelling of the wire-ring net and the spiral cables and successfully reproduced the non-linear force displacement response obtained in laboratory test by introducing additional parameters. An inverse optimization process based on the multi-island genetic algorithm is used for the determination of model parameters. For the chain-link nets, they developed a Finite Element model, and they also exploited a computational scheme relying on a





general contact algorithm to treat the complex contact interactions among chain-link elements and rockfall barriers components. Grassl et al. (2002) conducted a static test on a single ring in a ring net, in which the mechanical behaviour of

the ring net under the impact of rockfall was presented, the structure of passive rockfall barriers were simulated and analysed by self-programming and with different constraint forms being taken into account, and the results of numerical simulations and tests were compared. A numerical analysis tool was put forward to describe the energy dissipation performance of flexible rockfall barriers. In China, Liu et al. (2017) proposed the load model of the landslide pressure and provided a reference for the design of the passive rockfall barriers to resist the landslide; Wang Min et al. (2011;2012) studied the

mechanical properties and energy dissipation of ring net. The energy dissipation formula of a single ring under tension was deduced. Subsequently, the energy dissipation of the two ring nets and the influence of the boundary conditions on the energy dissipation performance of the ring net were presented. However, in the theoretical formula of a single ring, the determination of plastic hinge length is based on numerical simulation results, and there is no certain theoretical basis.

In summary, most current research results focus on the full-scale test of passive rockfall barriers and their overall

performance with simplified numerical simulation, whereas the basic research on ring net components is rare. Based on the current situation, the energy dissipation theoretical formula of a single ring under tension is derived, and its dynamic response, energy dissipation performance and failure mechanism of ring net under impact of rockfall are presented. It may provides a reference for a basic research and design of ring net components.

## 2 MECHANICAL PROPERTY ANALYSIS OF A SINGLE RING IN A RING NET

Depending on the type of connection between the rings, the ring net can be divided into the following two forms: a single ring connected with four peripheral rings, and a single ring connected with six peripheral rings (see Fig. 4).

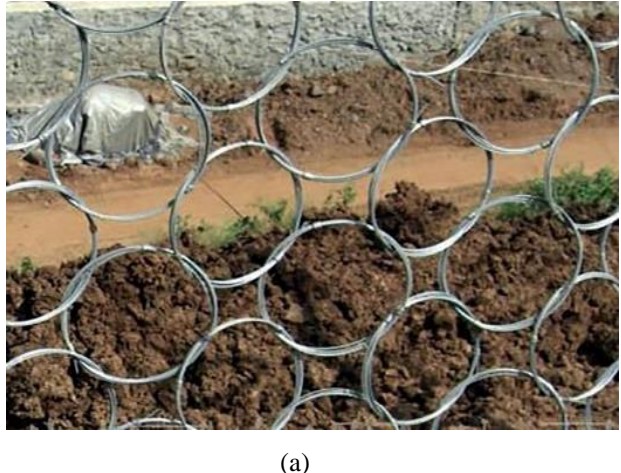
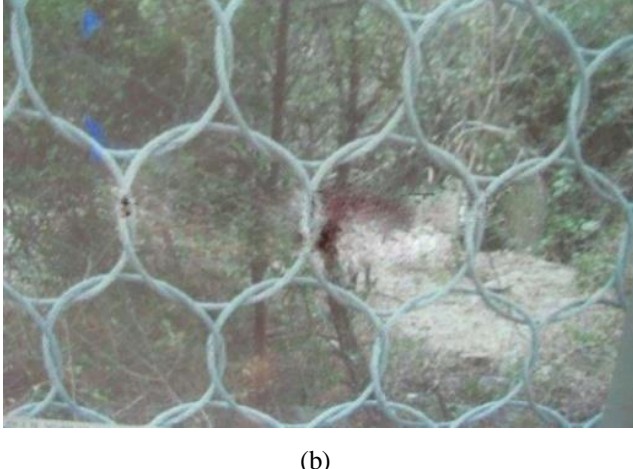

(a)                                                       (b)

**Fig. 4 Connection of rings. (a) 1 ring connected with 4 peripheral rings. (b) 1 ring connected with 6 peripheral rings**



Based on the energy absorption principle of materials and structures, the energy dissipation of a single ring under the tension of force can be divided into two parts—bending deformation energy dissipation and tensile deformation energy dissipation (Yu et al. 2006). In the bending deformation, the energy is required for the ring to be drawn into a rectangle or a hexagon. In the process of bending deformation, it is assumed that the ring stays the same length. In the tensile deformation, the ring produces tensile deformation to dissipate energy after pulled into rectangle or hexagon. The plastic strain of the ring

is much larger than elastic strain, from the energy dissipation point of view, the ring is assumed to be ideal rigid-plastic, the energy dissipation of tensile deformation occurs at discrete plastic hinges.

**2.1 Calculation of Theoretical Energy Dissipation of a Single Ring Under Two Tension**

**2.1.1 Calculation of bending deformation energy dissipation**

      When a single ring is subjected to two outward concentric forces that are equal and opposite, the plastic hinge position

corresponding to the maximum bending moment is moved with the increase of deflection, and the position of the plastic hinge moving with the deflection, which is called the moving hinge, with the undeformed arc segment and the deformed straight line always tangent at the plastic hinge B. In addition, the single ring stays the same length during the bending process. The deformation and force analysis are as shown in Fig. 5.

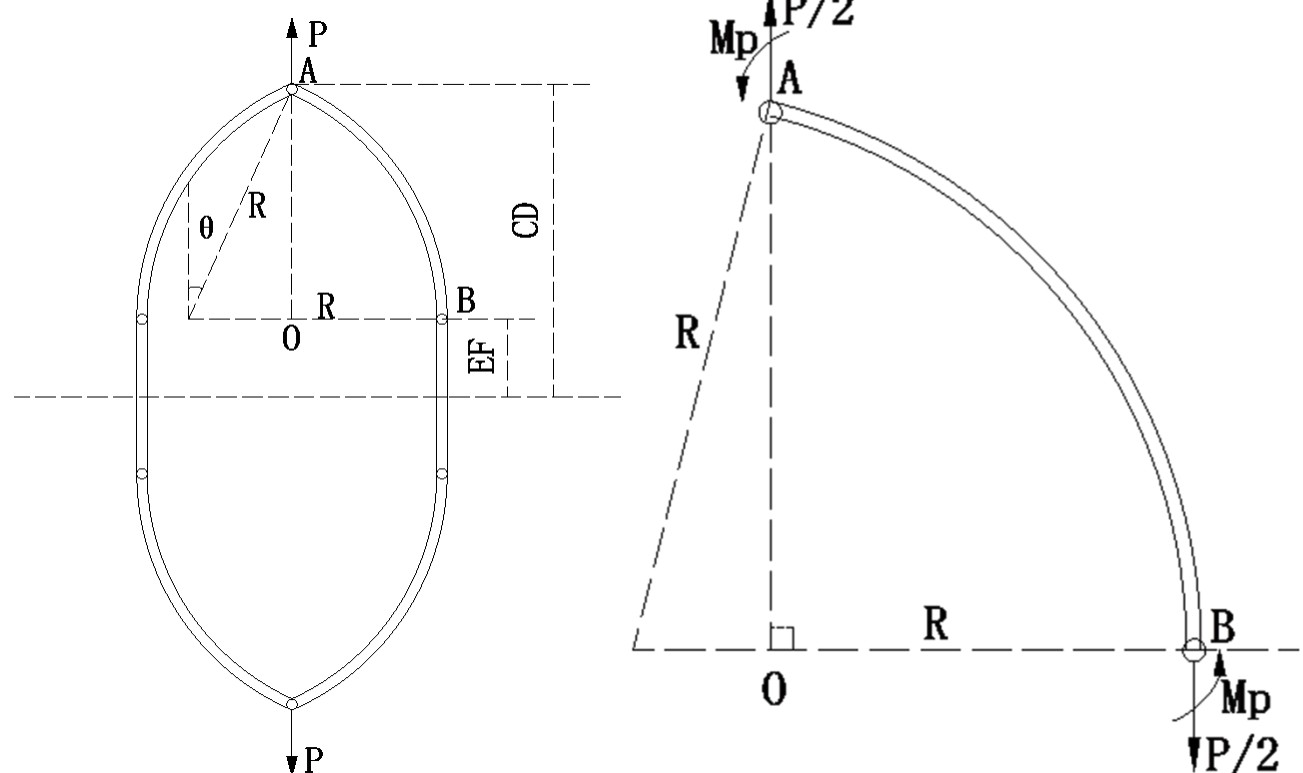





(a)                                                              (b)

**Fig. 5 Deformation and force analysis. (a) Deformation of the single ring under two-point tension. (b) Force analysis on a quarter of a single ring. (Note: It is assumed that in the process of damage, the two plastic hinges in the middle are separated into four, as shown by B in Fig. 5(a), from one to two)**

According to deformation assumptions, geometric relations and force analysis in Fig. 5, we can see a single ring under

tension at two points, the relationship between the force and displacement during bending deformation as shown in Eq. (1).

$$\begin{cases} \delta = 2R(\cos\theta + \theta - 1) \\ 2M_p = \frac{PR}{2}(1 - \sin\theta) \end{cases} \tag{1}$$

Where, $P$ is diametric tensile load; $R$ is the radius of the ring; $\theta$ is the angle variable; $\delta$ is the radial displacement

variation under tension load on diameters; $M_p$ is the plastic limit bending moment of the ring (Yu et al. 2006), $M_p = 4\sigma r^3/$

$3$, $\sigma$ is ring section yield stress, $r$ is the equivalent section radius of a single ring (Wang et al. 2012), $r = n^3 d/2$, $d$ is the

diameter of the ring section, $n$ is the number of coils of the ring.

According to the force-displacement curve in Eq. (1), the bending deformation energy dissipation $w_1$ of a single ring

under the tension of two points can be calculated as follows.

$$w_1 = \int_0^s P(\delta)d\delta \tag{2}$$

Where, $s$ is the maximum radial displacement under radial tension.

### 2.1.2 Calculation of tensile deformation energy dissipation

The influence of axial tension on yield is not considered in the above analysis, the single ring can dissipate energy by

tensile deformation after bending deformation. From the energy dissipation point of view, the energy dissipation of tensile

deformation occurs at discrete plastic hinges. For the ring under two forces, the tensile deformation energy dissipation $w_2$

can be calculated as following.

$$w_2 = 2M_p|\beta| \tag{3}$$

Where $\beta$ is the relative rotation angle of plastic hinge.

To sum up, the total energy dissipation $w$ of a single ring under the tension of two forces can be calculated as following,

$$w = w_1 + w_2 \tag{4}$$

### 2.2 Calculation of Theoretical Energy Dissipation of A Single Ring Under Four Tension

### 2.2.1 Calculation of bending deformation energy dissipation

Using the same research idea mentioned above, the structure and load of a single ring under the tension of four-point

tension are symmetry, the deformation and force analysis are as shown in Fig. 6.



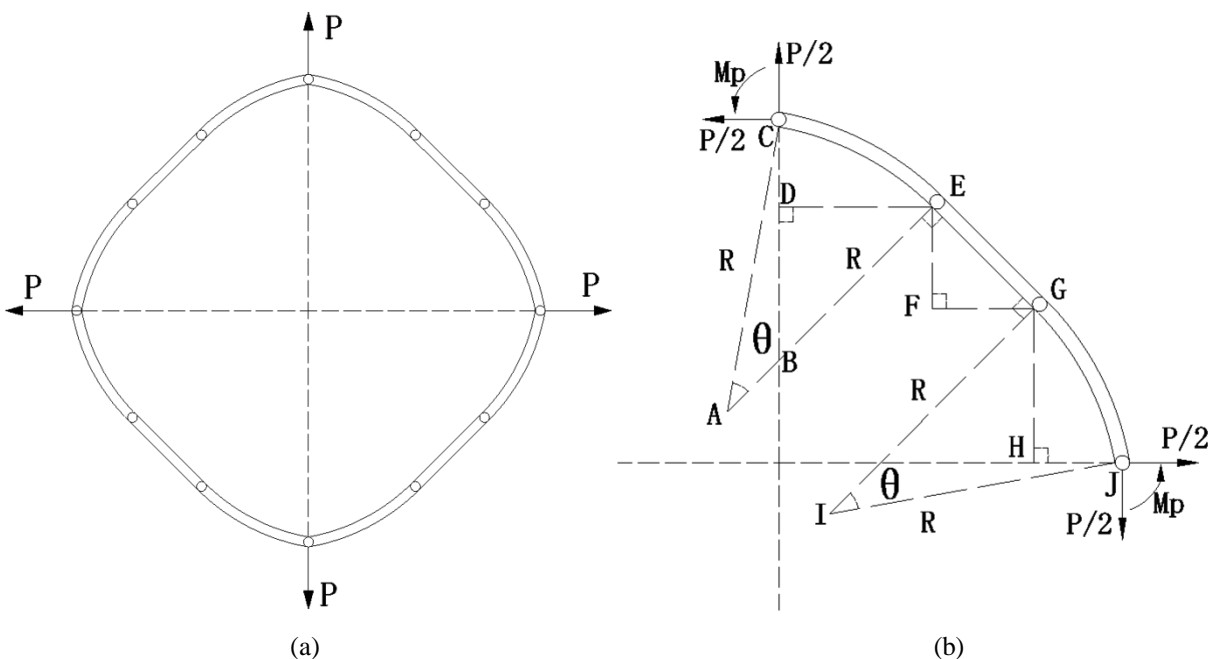

(a)                 (b)

**Fig. 6 Deformation and force analysis. (a) Deformation of the single ring under four-point tension. (b) Force analysis on a quarter of a single ring**

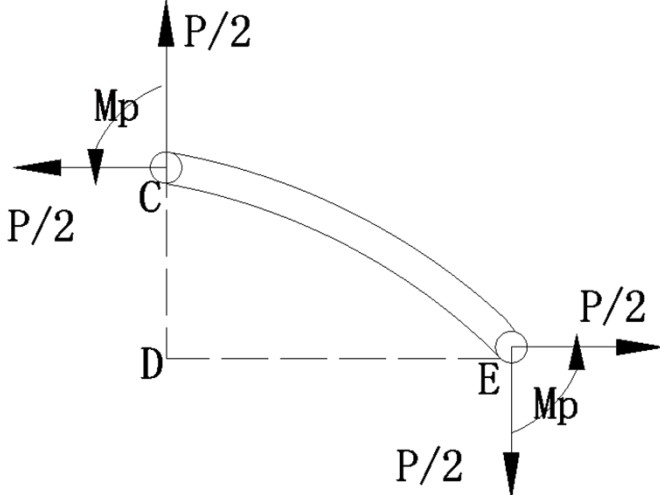

**Fig. 7 Force of the CE segment**

According to deformation assumptions, geometric relations and force analysis in Fig. 6, At the same time, we take the CE segment for stress analysis, as shown in Fig. 7, we can see a single ring under tension at four points, the relationship between the force and displacement during bending deformation as shown in Eq. (5).


$$\begin{cases} \delta = 2R(\sqrt{2}\sin\theta - \sqrt{2}\theta + \frac{\sqrt{2}\pi}{4} - 1) \\ 2M_p = \frac{\sqrt{2}}{2}PR(1 - \cos\theta) \end{cases} \quad (5)$$

Therefore, the bending deformation energy dissipation $w_1$ of a single ring under the tension of four points is calculated

from Eq. (5).

**2.2.2 Calculation of tensile deformation energy dissipation**

The energy can be dissipated through its tensile deformation before the ring become rectangle under four forces. Hence, $w_2$ can be calculated as follows.

$$w_2 = 4M_p|\beta| \quad (6)$$

So, the total energy dissipation $w$ of a single ring under the tension of four forces can be calculated according to Eq. (7).

$$w = w_1 + w_2 \quad (7)$$

**2.3 Calculation of Theoretical Energy Dissipation of a Single Ring Under Six Tension**

**2.3.1 Calculation of bending deformation energy dissipation**

Using the same research idea mentioned above, the symmetry of a single ring under the tension of six-point tension as

following. The deformation and force analysis are as shown in Fig. 8.

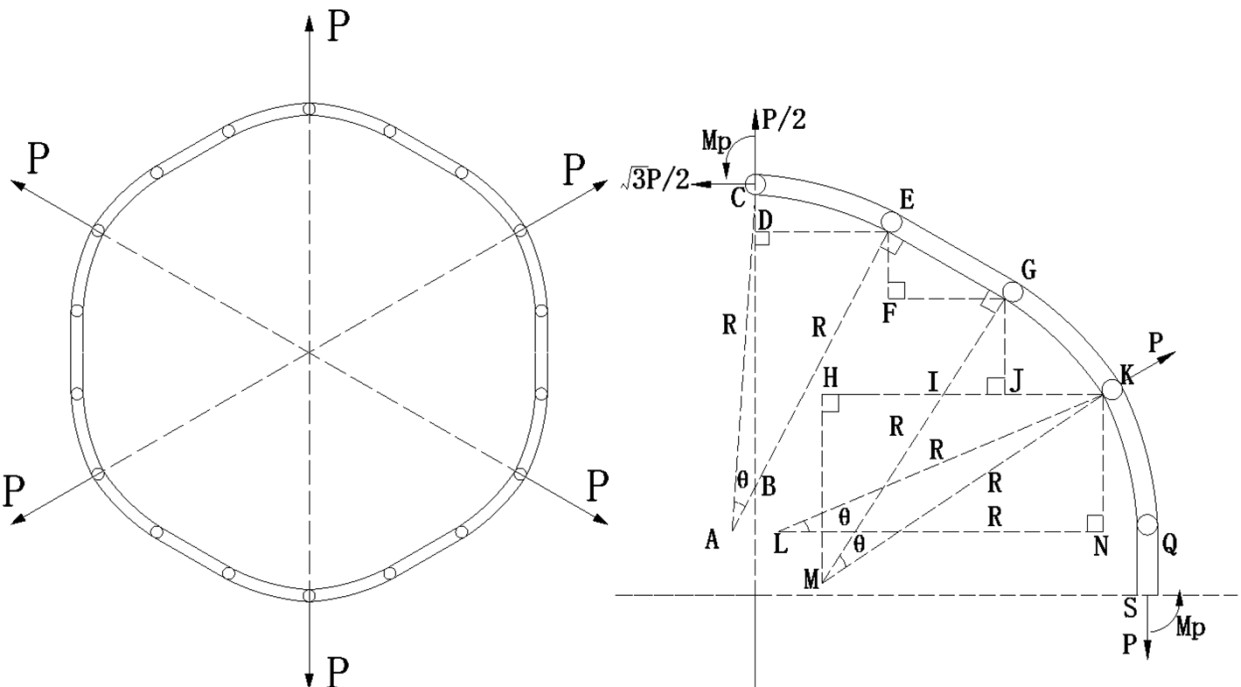


(a)                                                        (b)

**Fig. 8 Deformation and force analysis. (a) Deformation of the single ring under six-point tension. (b) Force analysis on a quarter of a single ring.**


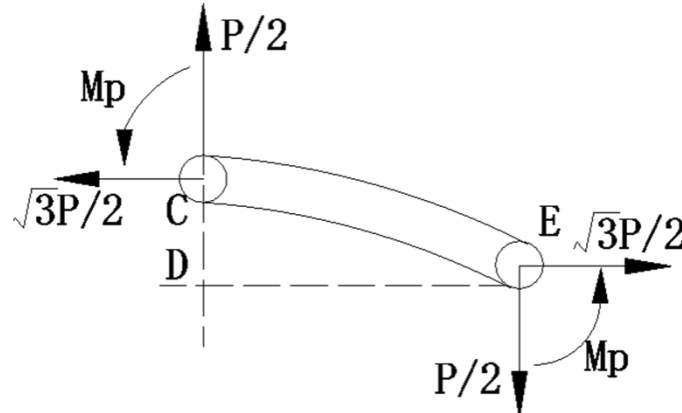

**Fig. 9 Force of the CE segment**

According to deformation assumptions, geometric relations and force analysis in Fig. 8, At the same time, we take the CE segment for stress analysis, as shown in Fig. 9, we can see a single ring under tension at six points, the relationship

between the force and displacement during bending deformation as shown in Eq. (8).

$$\begin{cases} \delta = 2R(2\sin\theta - 2\theta + \frac{\pi}{3} - 1) \\ 2M_p = PR(1 - \cos\theta) \end{cases} \tag{8}$$

Therefore, the bending deformation energy dissipation $w_1$ of a single ring under the tension of six points is calculated through Eq. (8).

### 2.3.2 Calculation of tensile deformation energy dissipation

The energy can be dissipated through its tensile deformation before the ring become rectangle under six forces. Hence, $w_2$ can be calculated as follows.

$$w_2 = 6M_p|\beta| \tag{9}$$

To sum up, the total energy dissipation $w$ of a single ring under the tension of six forces can be calculated according to Eq. (10).

$$w = w_1 + w_2 \tag{10}$$





## 2.4 Comparison and analysis with existing test results

Take the R7 / 3/300 ring as an example, a single ring with a diameter of 300mm is formed by a 3mm steel wire wound around 7 laps (Grassl et al. 2002). The results of energy dissipation formula are compared with the experimental results in existing literature. (Table 1).


**Table 1. A single ring energy consumption comparison**

|  | The results in literature | Theoretical results | Difference(%) |
|---|---|---|---|
| Two forces (Wang 2015) | 0.99 kJ | 1.01 kJ | 1.80 |
| Four forces (Wang et al. 2011) | 0.89 kJ | 0.95 kJ | 6.80 |
| Six forces | — | 0.90 kJ | — |

The errors of two and four forces are within the allowable range. Considering the fact that there is no introduction to the situation of a single ring under the tension of six forces. Therefore, because of the same basic principle used in deriving the energy dissipation of a single ring under force, it can indirectly prove the correctness of a single ring under the tension of six forces.

**3 Numerical Simulation of Ring Net under Impact of Rockfall and Comparative Analysis of Test Results**

A numerical model of the ring net for application in modelling flexible rockfall barriers is presented by ANSYS/LS-DYNA (ANSYS Inc.2006), the size of ring net is 3.9 m × 3.9 m. What's more, the ring net is fixed on all sides, with 180 rings while the ring type chosen is R7 / 3 / 300. In the experiment, the rockfall perpendicularly impacted the middle of the ring net, and the rockfall was a sphere with a mass of 830 kg and a density of 2600 kg/m³. The model material parameters

are as shown in Table 2.

**Table 2. Material parameters**

| Material | Modulus of elasticity | Density | Yield strength | Poisson's ratio | Ultimate strain |
|---|---|---|---|---|---|
| Wirerope | 1.77E+11 Pa | 7850 kg/m3 | 1.75E+09 Pa | 0.30 | 0.05 |
| Rockfall | 3.00E+10 Pa | 2600 kg/m3 | — | 0.30 | — |

Finite element model of the ring net under rockfall impact is established. The net size and constraint are the same as the literature (Grassl et al. 2002), and considering the influence of rockfall gravity, ignoring the relative slip between ring and ring. The rings are fixed, the impact energy of 24 kJ and 45 kJ is simulated by endowing different velocity of rockfall. The

finite element model is as shown in Fig. 10.



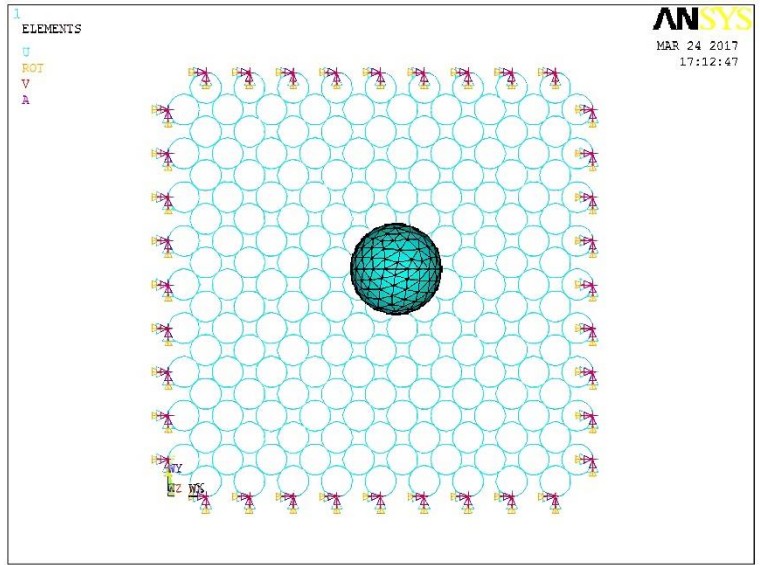

**Fig. 10 Finite element model**

The impact time is that the rockfall and the ring net begin to contact with the speed reduced to 0; The maximum displacement is that the rockfall and the ring net begin to contact with the speed reduced to 0.

The numerical simulation results of the impact energy of rockfall under 24 kJ and 45 kJ are compared with the experimental results in the literature (Grassl et al. 2002) as shown in Table 3.

**Table 3. Numerical simulation is compared with experimental results in literature .**

|  |  | Experimental result | Simulation | Difference(%) |
|---|---|---|---|---|
| maximum displacement (m) | 24kJ | 1.20 | 1.06 | 11.90 |
|  | 45kJ | 1.50 | 1.27 | 15.60 |
| maximum acceleration (m/s²) | 24kJ | 175.00 | 166.77 | 4.70 |
|  | 45kJ | 310.00 | 248.99 | 19.70 |
| impact time (s) | 24kJ | 0.19 | 0.17 | 12.10 |
|  | 45kJ | 0.15 | 0.15 | 0.00 |

From the comparison of data in Table 3, it can be seen that the numerical calculation method in this paper is similar to that of the experimental results under the impact of rockfall. The numerical analysis method proposed in this paper can be

used to simulate the dynamic process of ring net under the impact of rockfall, which provides a certain reference value for the overall analysis of passive rockfall barriers.



## 4 Dynamic Response and Energy Dissipation Analysis of Ring Net under Impact of Rockfall

### 4.1 Dynamic Response and Failure Mechanism of Ring Net

#### 4.1.1 The influence of different constraint forms on energy dissipation performance of ring net

According to the constraint of the boundary around the ring net, the boundary conditions can be divided into the three

forms, as shown in Figure 11.

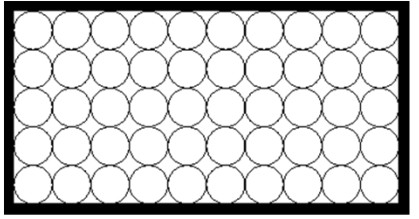 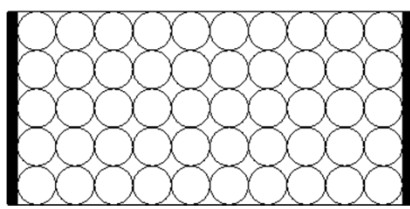 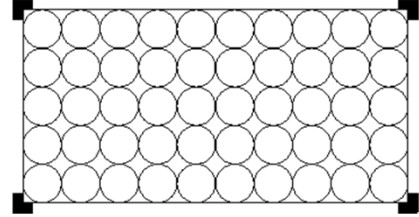

**Fig. 11 Three forms of ring net. Four sides are fixed (left). Both sides (opposite sides) are fixed (middle). Four corners are fixed(right)**

The finite element model of the ring net with 3.3 m × 6.6 m is established with ANSYS/LS-DYNA. The ring type is R7

/ 3 / 300, the ring net is surrounded by a 16 mm diameter support rope, the rockfall perpendicularly impacted the middle of

the ring net, and the rockfall was a sphere with a mass of 830 kg and a density of 2600 kg/m$^3$.

        The numerical simulation of three boundary conditions is carried out with a rockfall velocity of 7 m/s. and the

relationship between normal impact displacement and time under the three kinds of boundary conditions is as shown in Fig.

12, and the relationship between the falling rock energy change and time is as shown in Fig. 13.

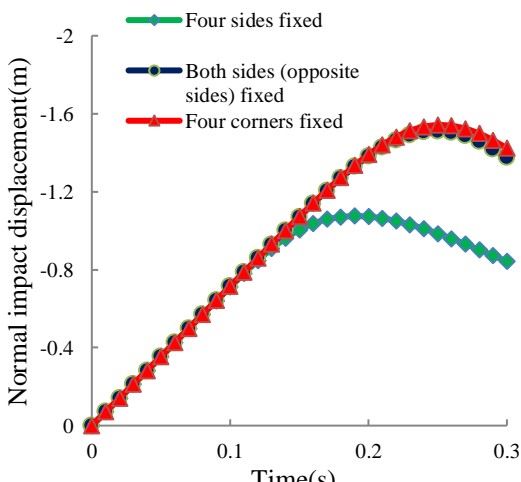

**Fig. 12 Relationship between normal impact displacement and time**





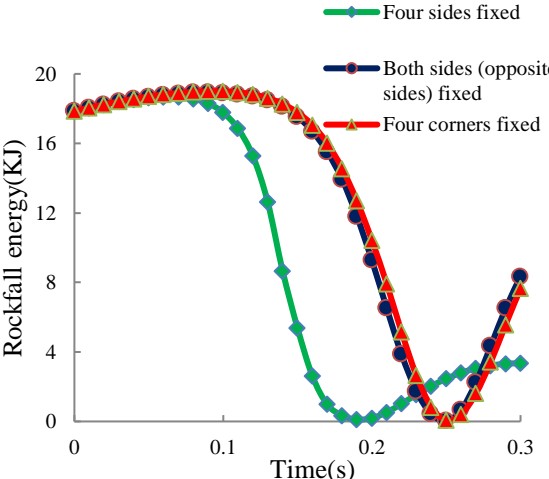

**Fig. 13 Relationship between rockfall energy and time**


With the release of peripheral constraints, the deformation capacity of the ring net is increased, the interaction time between the rockfall and the ring net is prolonged, and the energy dissipation performance of the ring net is improved. A further conclusion from the relationship between the energy variation and the time is that the release of the peripheral constraints will enhance the flexibility of the ring net and prolong the interaction time.

An analysis of the maximum energy dissipation of ring net under three different conditions are conducted. Assuming that the impact velocity of rockfall is $v_{lim}$, there is no damage to the ring net. If the impact velocity of rockfall is increased by 1 m/s, the impact velocity of rockfall is $v_{lim} + 1$, at this point, the ring net was damaged, according to the kinetic energy formula $E = \frac{1}{2}mv_{lim}^2$ the maximum energy dissipation of the ring net can be obtained.

The energy dissipation and damage form obtained by numerical simulation of ring net under three boundary conditions are as shown in Table 4. The initial state and failure of the ring net with four corners fixed are as shown in Fig. 14.


**Table 4. Energy dissipation and failure form of ring net under three boundary conditions**

| Boundary constraints form | four sides fixed | both sides fixed | four corners fixed |
|---|---|---|---|
| Damage form | ring broken | penetrating ring net | penetrating ring net |
| Maximum(kJ) | 33.62 | 50.22 | 59.76 |



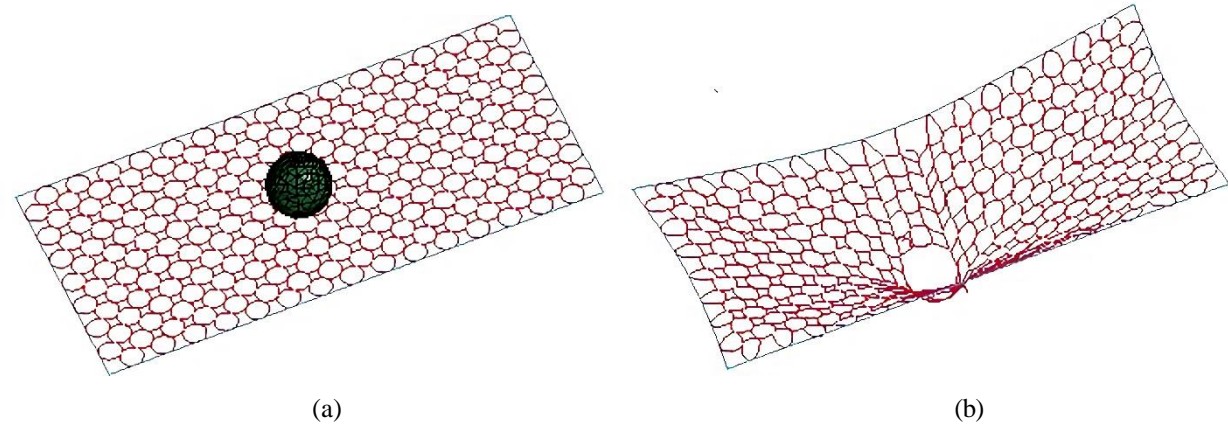

(a)                                      (b)

**Fig. 14 Initial state and failure of the ring net with four corners fixed.(a) Initial state. (b) Ultimate damage**

Therefore, for a single ring net, the release of peripheral constraints increases its deformation capacity and flexibility, prolonging the interaction time between the rockfall and the ring net. As a result, its energy dissipation capacity is improved greatly, which demonstrates the advantages of "overcoming hardness with softness".

### 4.1.2 The influence of rockfall impact angle on energy dissipation performance of ring net

In the analysis of passive rockfall barriers, it is generally assumed that the rockfall vertically impacts on the middle of them, but rockfall is usually dominated by rolling and bouncing (Wei et al. 2014). The rockfall will impact the ring net at a certain angle, which has different effects on the system. The boundary conditions used in the numerical simulation are as follows: with four corners fixed, the velocity of the rockfall is v=7 m/s, the impact time 0.6 s, the quality of rockfall 830 kg, and the rockfall density 2600 kg/m$^3$. According to the actual application in the project, the impact angle is selected from the

following five situations: 0 °, 15 °, 30 °, 45 ° and 60 °. The dynamic response of ring net with different impact angles are analyzed by decomposition of velocity. The impact model is as shown in Fig. 15 and the corresponding finite element model as shown in Fig. 16.




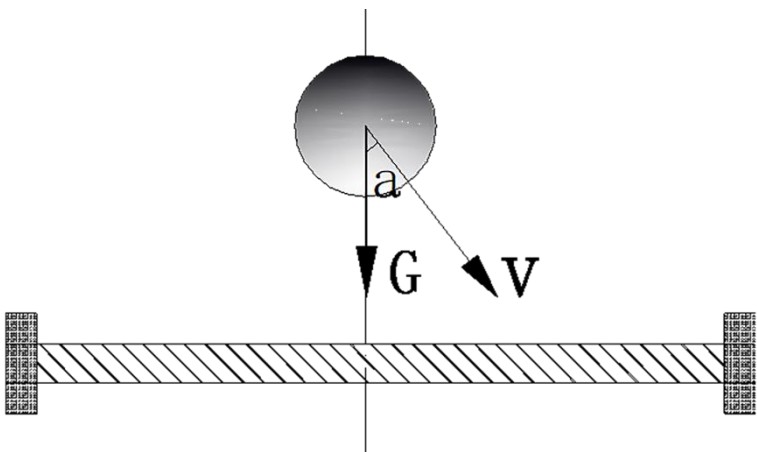

**Fig. 15 Impact model**

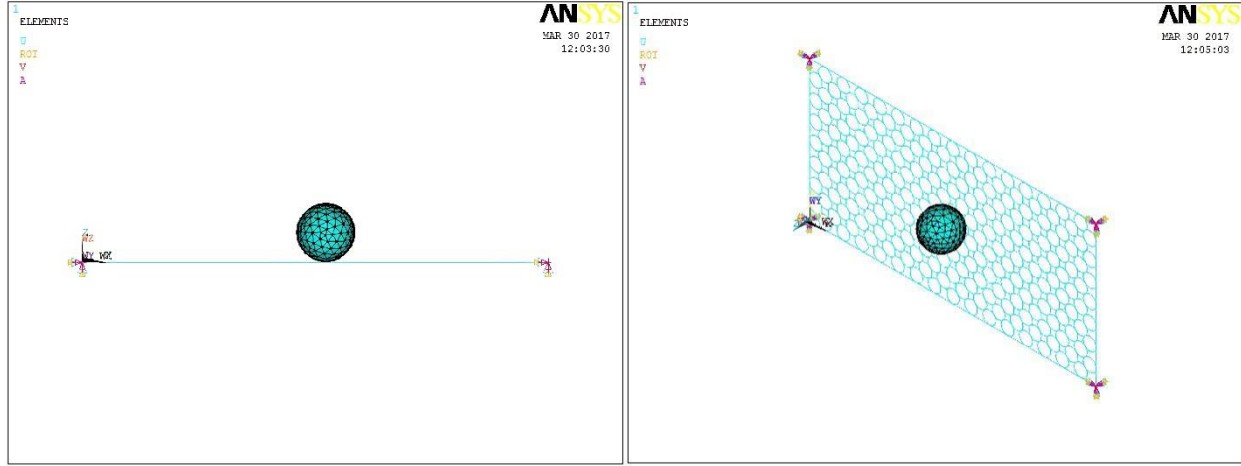


**Fig. 16 Finite element model**

The maximum velocity of rockfall and the destruction method of ring net are analyzed under different impact angles (see Table 5.).

**Table 5. Ring net maximum dynamic response**

| Impact angle (°) | Critical speed (m/s) | Rockfall kinetic energy (kJ) | Destruction method |
|---|---|---|---|
| 0 | 12 | 59.76 | rockfall penetrating ring net |
| 15 | 10 | 41.50 | the fracture of the central ring at the connection of the lower support rope |
| 30 | 10 | 41.50 | the fracture of the central ring at the connection of the lower support rope |
| 45 | 9 | 33.62 | the fracture of the central ring at the |



| | | | connection of the lower support rope |
|---|---|---|---|
| 60 | 8 | 26.56 | the fracture of the central ring at the connection of the lower support rope |

It can be seen from the above table that for the same ring net, the critical velocity and kinetic energy of rockfall decreases with the increase of impact angle. When the impact angle is 0 °, the kinetic energy of rockfall is 59.76 KJ, and when the impact angle is 60 °, the kinetic energy of rockfall is 26.56 KJ, the difference is 2.25 times. With the increase of the impact angle, the destruction method of the ring net is the fracture of the central ring at the connection of the lower support rope.

Because the trajectories of rockfalls do not always strike the ring net vertically, there are several additional measures that should be taken to deal with the situation in the practical application of the overall barriers, for example: adding brake rings in the support rope to increase the deformation of the passive rockfall barriers and prolong the interaction time; adding the cross-sectional size of the ring connecting with the supporting rope; and adding the cross-section of the supporting rope, etc.

**4.2 Energy dissipation Analysis of Ring Net**

According to the energy dissipation theoretical formulas of a single ring under the tension of two forces, four forces and six forces, the energy dissipation of ring net at four corners fixation is studied. The discussion is based on the connection between a single ring and four rings.

Model overview: the ring type is R7/3/300, the impact time 0.3 s, rockfall mass 830 Kg, rockfall density 2600 kg / $m^3$, 270 the boundary condition fixed at four corners, the middle position of the ring net impacted vertically by rockfall, and the direction of rockfall gravity and velocity are the same. From the Eq. (7), we can see the total energy dissipation $w$ of a single ring under the tension of four points is 0.95kJ, bending deformation energy dissipation $w_1$ is 0.58 kJ.

By changing the number of rings in the transverse and longitudinal directions, the size of the ring net is changed. The maximum number of rings in the transverse is $n$, and the maximum number of rings in the longitudinal direction is $m$ (see 275 Fig. 17).

The total number N of rings included in each ring net is: $N = 2mn - m - n$, and the horizontal dimension X or Y of the ring net is: $X = 2\sqrt{2}R \times (n - 1) + 2R$, or $Y = 2\sqrt{2}R \times (m - 1) + 2R$.



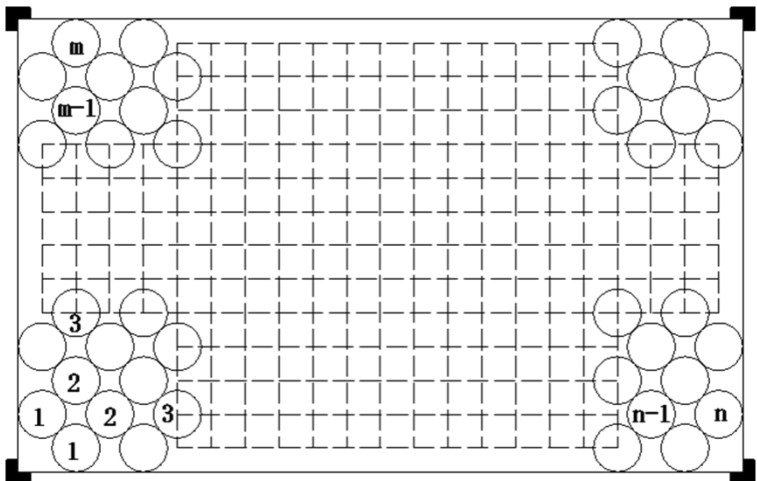

**Fig. 17 Ring net**

Now, using a rectangular ring net (n = 12, m = 6, shown in Fig. 18) as an example for verification the theoretical energy dissipation formulas, The maximum speed of rockfall is 10m/s. The maximum kinetic energy of rockfall is 41.5kJ.

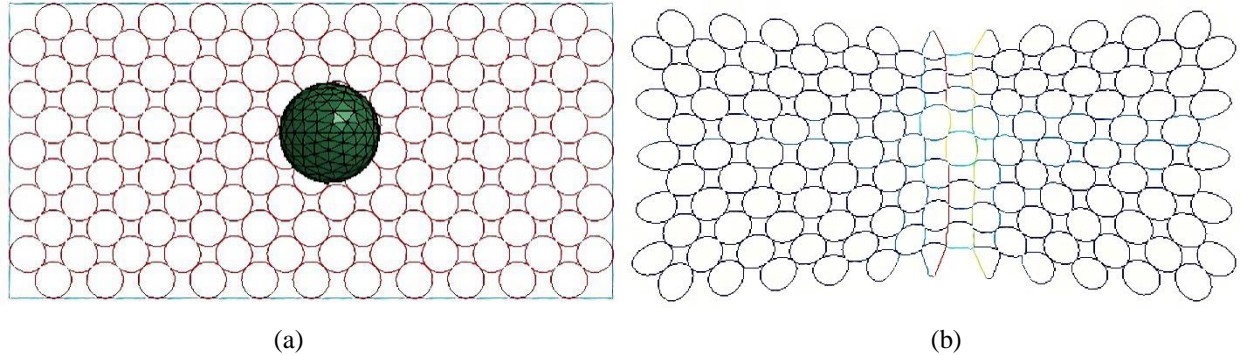

(a)                                                          (b)

**Fig. 18 n = 12, m = 6, a finite element model. (a) Ring net initial state. (b) Maximum axial force distribution of ring net**

It can be seen that the deformation area of the ring net is a cross shape. Assuming that the rings in the vertical strip region all reach plastic deformation, the bending deformation energy of the rings in the transverse strip region thus calculated reaches the maximum. A comparison of the two energy dissipation results could be conducted so as to validate the plausibility of the hypothesis. The cross region is as shown in Fig. 19.



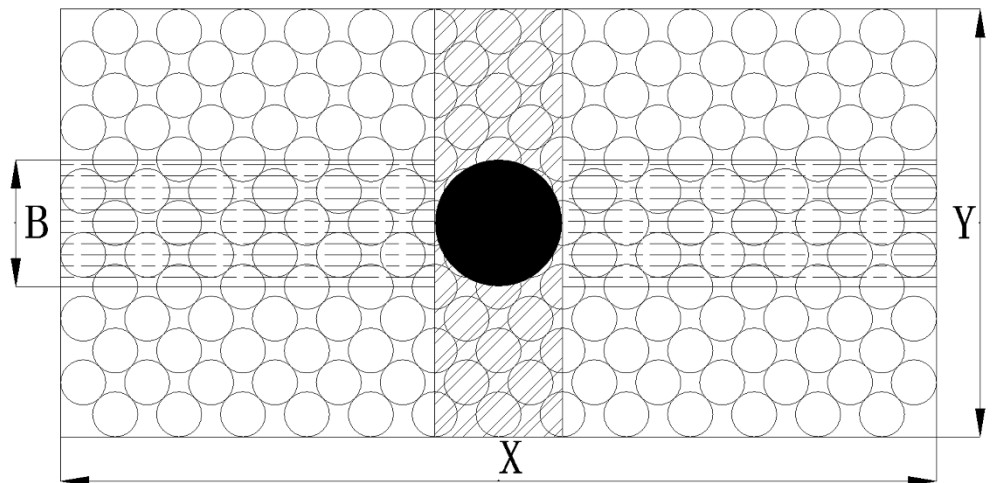

**Fig. 19 Cross region of ring net (note: the rings in the transverse strip region only calculate its bending deformation energy dissipation energy, and the rings in the vertical strip region calculate bending deformation energy dissipation and tensile deformation energy dissipation)**

Based on Fig. 19, it is calculated that: Area S of ring net: $S = XY$;Area $S_1$ of vertical strip region:  $S_1 = BY$   ($B$ is the width of the cross-shaped area, that is, the diameter of the rockfall); Area $S_2$ of transverse strip region:   $S_2 = (X - B)B$;

Number $N_1$ of rings in the vertical strip region:$N_1 = N\frac{S_1}{S}$; Number $N_2$ of rings in the transverse strip region: $N_2 = N\frac{S_2}{S}$; The energy dissipation $W_1$ of the ring in the vertical strip region:  $W_1 = N_1 w$;The energy dissipation $W_2$ of the ring in the transverse strip region:  $W_2 = N_2 w_1$.

The energy dissipation $W$ of the ring in the cross region is the energy dissipation of the ring net:

$$W = W_1 + W_2 = N_1 w + N_2 w_1 \tag{11}$$

In order to verify the correctness of the above Eq. (11), the result of n = 12, m = 6 is substituted into the Eq. (11):

$$W = W_1 + W_2 = N_1 w + N_2 w_1 = 22 \times 0.95 + 36 \times 0.58 = 41.78 \text{kJ}$$

The relative error between numerical simulation and theoretical calculation is:

$$\frac{41.78 - 41.5}{41.5} = 0.67\%$$

The error is small, so for different size of the ring net, this equation is still applicable, respectively, and then take n =

14, m = 7 and n = 16, m = 8 for analysis (see Fig. 20, Fig. 21).





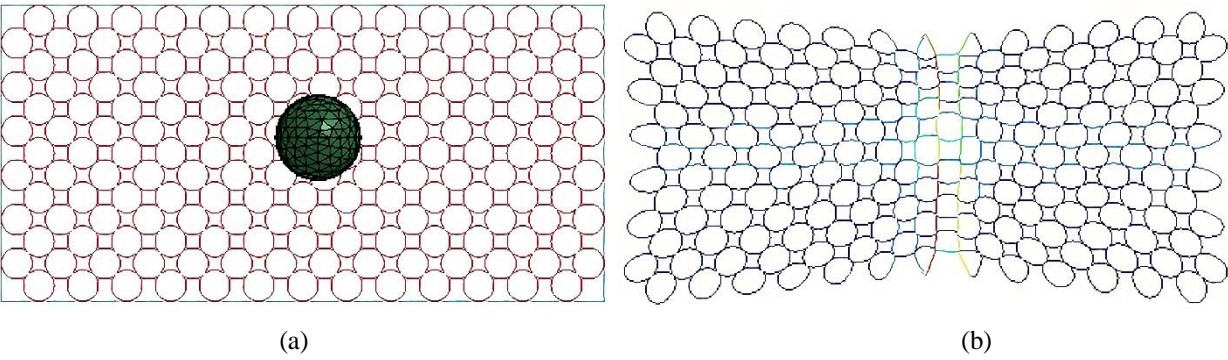

(a)                                                             (b)

**Fig. 20  n = 14, m = 7, a finite element model. (a) Ring net initial state. (b) Maximum axial force distribution of ring net**

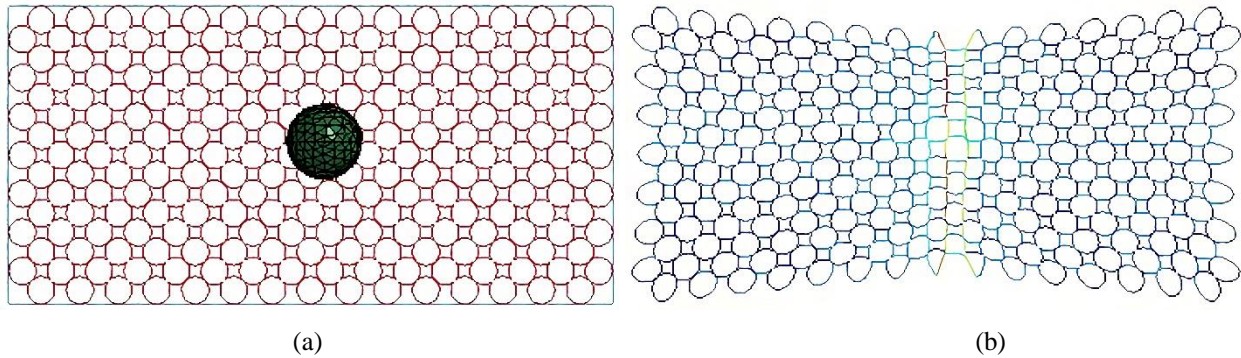

(a)                                                             (b)

**Fig. 21  n = 16, m = 8, a finite element model. (a) Ring net initial state. (b) Maximum axial force distribution of ring net**

The relative errors of results between theoretical calculation and numerical simulation are as shown in Table 6.

**Table 6. Relative error of energy dissipation of rectangular ring net**

| Number of ring(n×m) | The maximum speed of rockfall(m/s) | Simulation(kJ) | Calculation (kJ) | Difference(%) |
|---|---|---|---|---|
| 12×6 | 10 | 41.50 | 41.79 | 0.71 |
| 14×7 | 10.5 | 45.75 | 48.71 | 6.47 |
| 16×8 | 12 | 59.76 | 58.42 | 2.24 |

The correctness of these equations is verified by this comprehensive analysis of the three conditions, that is, for the rectangular ring net, once the size of the ring net is determined, it can be used to calculate and estimate its energy dissipation approximately.




## 5 Conclusions

(1) The energy dissipation capacity of a single ring under application of two point, four point and six point tensile loading are studied, the correctness of the theoretical results is verified by experiments.

(2) A numerical model of the ring net for application in modelling flexible barriers is presented. The numerical simulation results are compared with the existing experimental results, and a reasonable numerical model is provided for the dynamic study of the ring net.

(3) The effects of boundary condition and rockfall impact angle on the energy dissipation of ring net are studied by numerical simulation. With the release of peripheral constraints, the deformation capacity and flexibility of the ring net is increased, and the energy dissipation performance of the ring net is improved. As the impact angle increases, the impact energy of the rock on the ring net will experience a gradual decline.

(4) The theoretical energy dissipation capacity of ring net under the impact of rockfall is given quantitatively, which provides a theoretical basis for the design of the passive rockfall barriers.

## Acknowledgements

All authors would like to express their gratitude to the support of Natural Science Foundation of China (51778538).

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
