# Peer review of "Study on Mechanical Properties and Dissipation Capacity of Ring Net in Passive Rockfall Barriers"

_Natural Hazards and Earth System Sciences, 2018_

## Referee Comment (RC1) · Anonymous Referee #1 · 29 Jun 2018

The article handles the mechanical performance of net rings within rockfall protection nets and their numerical simulation. It therefore queues into the series of existing research on this type of nets. Based on the fact that already a lot of research exists on the ring nets I would have expected more comparison with existing research.

Specific comments: - The article uses the expression "brake rings". Please, be aware that the role of the handles brake rings are the so-called energy absorbing elements. There are not only brake rings around to perform this task depending on the manufacturer of a rockfall protection system. Therefore, I would change from "brake rings" to "energy absorbing elements".

- You are considering 2-, 4- and 6-point bending of the net rings. However, corner and edge rings in the 4-fold-ring nets are connected to 3 points. Your mechanical numerical

analyses should include this setup.

- There is much more research on ring nets around as stated so far in the references. E.g. the works of Nicot in the late nineties are left our completely (use scholar.google and search for "nicot rockfall" for suitable references).

- P2L32: There are much more net types around. Have a check on the products of Trumer, Jakob, Isofer, etc.!

- P2L34: Please, add some references for destructed ring nets!

- P2L35: "ring net the" –> "ring net. The"

- P3L41: What does "foreign" mean? Your publication is meant to be read world wide. If you are interested to publish only for China then "foreign" might be ok.

- P3L43: "characterizing" –> "characterize"

- P3L49-51: This sentence does not fit in here. "Tecco" is no ring-net. Further it hasn't been described before.

- P3L52: "33% so" –> "33%. So"

- P3L62: Chain-link nets were not described before.

- P6L106: If you directly add theta with cos(theta) you should describe in which unit theta has to be used.

- P6L109: Grassl (2002) reports a different equivalent section radius. Please discuss. (Grassl, H. G. (2002). Experimentelle und numerische Modellierung des dynamischen Trag- und Verformungsverhaltens von hochflexiblen Schutzsystemen gegen Steinschlag (Doctoral dissertation, Ph. D. Diss. Swiss Federal Inst. of Technology Zurich, Switzerland).

-P6: If you have a plastically deformed net ring (2-, 3- or 4 point tension) and you cut it at one place completely through the "ring" shape gets lost and the it snaps inwards.

This shows that the deformed ring stores a lot of elastic energy. Please, quantify and discuss this part.

- P6L124: add "point" between "four" and "tension"

- P8L139: Does the bending deformation energy dissipation depend on the bending radius?

- P8L147: Please compare with the analytical solution of Nicot (see above).

- P8L175: Please, compare with the results of Grassl (2002).

- P8L181: "presented" –> "realized" ?

- P11Table3: Please, explain the displacement measurements. Are they including the static sag? Are they separated from the static sag? How has the static sag been treated in simulation?

- P11Table3: What is the maximum energy capacity of this setup? Compare it with the results of Grassl(2002).

-P12L204: This section has a fundamental mismatch. If a ring net is attached to a circumferential rope, the rings can slide along the rope.This significantly changes the load bearing capacity of a ring net. Please, compare, discuss, adjust....

-P12Fig.11: Please, arrange the drawn rings as they are arranged in simulation.

- P13L231: Do you have comparable results from experiments?

- P14L244/245: Add "alpha" somewhere

-P14Fig14&Fig17: Please, be aware that manufacturer uses the ring net for typical barrier panels with rings in the four courners! This changes the load bearing capacity. Further, numbering of rows in Fig. 17 is not congruent with manufacturers numbering!

- P15Fig.15: Change "a" to "alpha".

[Figure]

- P15Fig.16: The right figure gives the impression of a vertical barrier with no "g" acting on the net. Please, choose a different viewing angle.

- P15Table5: The desctruction method is repeated identically four times. Please, adjust table to avoid this repitition.

- P16L260: "vertically" –> "orthogonally"?

- P19Table6: "maximum" –> "impact"

- P20L334: Please, sort the references alphabetically or use numbering if you want to keep the current order.

- P20L347: ".,:" –> ":"

- P21L361: "DANY" –> "DYNA"

―――――――――――――――――――――

---

## Author Comment (AC1) · 30 Jul 2018

Dear Referee, Thank you for your comments concerning our manuscript entitled "Study on Mechanical Properties and Dissipation Capacity of Ring Net in Passive Rockfall Barriers" (Manuscript Number: nhess-2018-76). Those comments are all valuable and very helpful for revising and improving our paper, as well as the important guiding significance to our researches. We have studied comments carefully and have made correction. We hope these revisions will meet with approval.The attachment contains the revised manuscript, response and the marked-up manuscript version.The main corrections in the paper and the responds to your comments are as follows: - The article uses the expression "brake rings". Please, be aware that the role of the handles brake rings are the so-called energy absorbing elements. There are not only brake

rings around to perform this task depending on the manufacturer of a rockfall protection system. Therefore, I would change from "brake rings" to" energy absorbing elements". Response: Thank you for your good advice. We think it is appropriate to change from "brake rings" to" energy absorbing elements". We have revised it in the paper.

- You are considering 2-, 4- and 6-point bending of the net rings. However, corner and edge rings in the 4-fold-ring nets are connected to 3 points. Your mechanical numerical analyses should include this setup. Response: Thank you for your good advice. The corner and edge rings in the 4-fold-ring nets are indeed connected to 3 points. This is the problem we need to continue researching next. However, this paper mainly studies the mechanical properties and dissipation capacity of the ring net. The proportion of the 3-point connecting ring is relatively small, which has little effect on the energy consumption of the whole net, so we think the mechanical numerical analyses can exclude this setup.

- There is much more research on ring nets around as stated so far in the references. E.g. the works of Nicot in the late nineties are left our completely (use scholar.google and search for "nicot rockfall" for suitable references). Response: Thank you for your good advice. Nicot et al. (2012) undertaken to analyse the mechanical behaviour of rockfall restraining structures. A spatial description of the net is proposed, and a constitutive modelling in finite strains is presented. The proposed modelling is based on both the experimental and analytical approaches. But the main feature of this modelling is that the local behaviour of the net can be described in the usual framework of elasto-plasticity. We have added their research to the paper.

- P2L32: There are much more net types around. Have a check on the products of Trumer, Jakob, Isofer, etc.! Response: Thank you for your good advice. the widely-used metal flexible nets are usually presented in the following forms: ring nets, rhombic nets, chain-link wire nets and omega nets. We have supplemented it in the paper.

- P2L34: Please, add some references for destructed ring nets! Response: Thank you

for your good advice. We have added some references for destructed ring nets. Liu, C.Q., Wei, X.D., Lu, Z., Wu, H.D., Yang, Y.L., Chen, L.Y.: Studies on passive flexible protection to resist landslides caused by the May 12, 2008, Wenchuan earthquake, Struct. Design Tall Spec. Build., 26(11), 2017. DOI: 10.1002/tal.1372. Wendeler, C., Volkwein, A.: Laboratory tests for the optimization of mesh size for flexible debris-flow barriers, Nat. Hazards Earth Syst. Sci., 15(12):2597–2604, 2015. Canelli, L.; Ferrero, A. M.; Migliazza, M.: Debris flow risk mitigation by the means of rigid and flexible barriers – experimental tests and impact analysis, Nat. Hazards Earth Syst. Sci., 12(5):1693–1699, 2012. Gianfreda, F.; Mastronuzzi, G.; Sanso, P.: Impact of historical tsunamis on a sandy coastal barrier: an example from the northern Gargano coast,southern Italy, Nat. Hazards Earth Syst. Sci., 1 (4), 2001.

- P2L35: "ring net the" –> "ring net. The" Response: Thank you for your good advice. We have revised it in the paper.

- P3L41: What does "foreign" mean? Your publication is meant to be read world wide. If you are interested to publish only for China then "foreign" might be ok. Response: Thank you for your good advice. We have revised it in the paper. The modification is as follows: Currently, it is true that some researchers have made great achievements in the structure of passive rockfall barriers. Although there was barely no difference all their test studies, the research on mechanics is still inadequate.

- P3L43: "characterizing" –> "characterize" Response: Thank you for your good advice. We have revised it in the paper.

- P3L49-51: This sentence does not fit in here. "Tecco" is no ring-net. Further it hasn't been described before. Response: Thank you for your good advice. certainly, "Tecco" is no ring-net. We have deleted this reference and added the following references: Escallon et al. (2013) presents quasi-static and impact explicit FE simulation results of wire-ring net tests using an approach which relies on the general contact algorithm available in the FE code Abaqus. This approach allows a better description of the

physics involved in impact problems related to rock falls. The model accounts for many complex physical processes: high-speed impact, contact with sliding friction, damage initiation and evolution, and strain-rate dependent material behavior.

- P3L52: "33% so" –> "33%. So" Response: Thank you for your good advice. We have deleted this reference.

- P3L62: Chain-link nets were not described before. Response: Thank you for your good advice. We have described Chain-link nets in Fig.3.

- P6L106: If you directly add theta with cos(theta) you should describe in which unit theta has to be used. Response: Thank you for your good advice. $\theta$ is the angle variable, according to deformation assumptions, geometric relations and force analysis in Fig.5, we can calculate equation 1.

- P6L109: Grassl (2002) reports a different equivalent section radius. Please discuss. (Grassl, H. G. (2002). Experimentelle und numerische Modellierung des dynamischen Trag- und Verformungsverhaltens von hochflexiblen Schutzsystemen gegen Stein-schlag (Doctoral dissertation, Ph. D. Diss. Swiss Federal Inst. of Technology Zurich, Switzerland). Response: Grassl hans gerhard (2002) conducts dimensional analysis of the ring-net barriers components in the application empirical design procedures, and full-scale tests were performed using Single and three-span net configurations, net deformations and cable forces over time were measured. In parallel to the experimental research, a simplified explicit finite element program was developed. This program was coupled with a structural reliability program and used to analyse the reliability of the protection Systems.

-P6: If you have a plastically deformed net ring (2-, 3- or 4 point tension) and you cut it at one place completely through the "ring" shape gets lost and the it snaps inwards. This shows that the deformed ring stores a lot of elastic energy. Please, quantify and discuss this part. Response: Thank you for your good advice. However, the plastic ring is cut off, and the energy produced by rebound is not within the scope of our study,

which is what we need to further study in the future. - P6L124: add "point" between "four" and "tension" Response: Thank you for your good advice. We have revised it in the paper.

- P8L139: Does the bending deformation energy dissipation depend on the bending radius? Response: The bending deformation energy dissipation does not depend on the bending radius.it depends on the diametric tensile load P, the radius of the ring R, the angle variable $\theta$, the radial displacement variation under tension load on diameters $\delta$; M_p is the plastic limit bending moment of the ring.

- P8L147: Please compare with the analytical solution of Nicot (see above). Response: Nicot's research is mainly focused on ASM nets, which is different from the one used in our research, so there is no comparability.

- P8L175: Please, compare with the results of Grassl (2002). Response: According to the results of Grassl (2002), the energy consumption of a single ring under four-point tension is about 10kJ, which is quite different from the results we calculated. However, according to the results of wang (Wang, M.: Rockfall impact protection system, Ph. D. Diss, Chongqing, Logistical Engineering University, 2011.), the energy consumption is 0.89kJ. That's pretty close to our calculation, this is an interesting question and we need to study it further.

- P8L181: "presented" –> "realized" ? Response: Yes. "presented" –> "realized".

- P11Table3: Please, explain the displacement measurements. Are they including the static sag? Are they separated from the static sag? How has the static sag been treated in simulation? Response: The results of the experiment are based on the work of Grassl (2002). The influence of static sag is not considered in simulation, some experiment results in table 3 are quite different from the numerical simulation results, and the failure to consider static sag in simulation is one of the important reasons.

- P11Table3: What is the maximum energy capacity of this setup? Compare it with the

results of Grassl(2002). Response: The maximum energy capacity is the energy of falling stone, which is 24kJ and 45 kJ respectively.

-P12L204: This section has a fundamental mismatch. If a ring net is attached to a circumferential rope, the rings can slide along the rope. This significantly changes the load bearing capacity of a ring net. Please, compare, discuss, adjust.... Response: Thank you for your good advice. In practice, the ring net is attached to a circumferential rope, the rings can slide along the rope, but in this paper, we simplify the boundary condition and simplify them into three forms: four-sided fixation, two-sided fixation and four-corners fixation. The boundary condition of sliding connection is a subject that needs further study.

-P12Fig.11: Please, arrange the drawn rings as they are arranged in simulation. Response: Thank you for your good advice. We have arranged it in the paper.

- P13L231: Do you have comparable results from experiments? Response: This is the result of our numerical simulation, and it is too late to do the experiment comparison, but it can provide a certain reference for the boundary conditions of the ring nets.

- P14L244/245: Add "alpha" somewhere Response: Thank you for your good advice. We have revised it in the paper.

-P14Fig14&Fig17: Please, be aware that manufacturer uses the ring net for typical barrier panels with rings in the four courners! This changes the load bearing capacity. Further, numbering of rows in Fig. 17 is not congruent with manufacturers numbering! Response: Thank you for your good advice. In the course of our study, the ring network was simplified, and in order to make the research universal, the research did not follow the manufacturer's typical ring network type.

- P15Fig.15: Change "a" to "alpha". Response: Thank you for your good advice. We have revised it in the paper.

- P15Fig.16: The right figure gives the impression of a vertical barrier with no "g" acting

on the net. Please, choose a different viewing angle. Response: Thank you for your good advice. We have revised it in the paper.

- P15Table5: The desctruction method is repeated identically four times. Please, adjust table to avoid this repitition. Response: Thank you for your good advice. We have summarized the ways of destruction in the paper, combining four times into one.

- P16L260: "vertically" –> "orthogonally"? Response: Yes, vertically" –> "orthogonally.

- P19Table6: "maximum" –> "impact" Response: Yes, The maximum speed of rockfall –> The maximum impact speed of rockfall.

- P20L334: Please, sort the references alphabetically or use numbering if you want to keep the current order. Response: Thank you for your good advice. We have rearranged the references alphabetically.

- P20L347: ".,:" –> ".:" Response: Thank you for your good advice. We have revised it in the paper.

- P21L361: "DANY" –> "DYNA" Response: Thank you for your good advice. We have revised it in the paper.

Please also note the supplement to this comment:
https://www.nat-hazards-earth-syst-sci-discuss.net/nhess-2018-76/nhess-2018-76-AC1-supplement.zip

---

## Referee Comment (RC2) · Anonymous Referee #1 · 27 Aug 2018

Dear authors,

thank you for considering my previous comments. Most of them were considered satisfyinlgy. Let me allow to comment some of your revisions. My comments are related to the linenumbers of the version including all tracked changes.

L44ff: Your judgement on the existing research is to general.

- There is still more existing research to include in this overview section. Please, intensify your literatur research. It is not my part to list all the single research works that exist since the late nineties. For example, the original works of Nicot were not published in 2012 but much earlier in the late nineties and at the beginning of the millenium. Or Volkwein (2004) setup a special discrete element for net rings ("Volkwein,

[Figure]

A. (2004). Numerische simulation von flexiblen steinschlagschutzsystemen (No. 289). vdf Hochschulverlag AG."). And much more publications exist.....

- You wrote that there is barely no differenc ein the existing research. Please, explain both what is the same between these researches and what is the difference of your research to the existing ones. From my point of view, your research is pretty much the same.

L46: After reading this sentecne I would expect that your article finally brings the adequate mechanics. However, it still lacks a lot (three point tension, comparison with the analytical solution of Nicot (1998/99) etc.).

L75: "Grassl hans gerhard" –> "Grassl"

L108: Insert "Point" between "Two Tension" (same as in L140).

Section2: In your reply to my previous comments regarding 3-point-tensioned rings you stated that the influence of these rings is marginal. However, your calculation examples contain 20-33% of rings that are connected at three and not four points. I would estimated that this number is not small. Please quantify the influence and error induced by this assumption.

Fig.5: Where are the points CDEF? Are they need anyhow? Do you need ER & CD in this figure?

L122: Which unit has to be taken for theta? Is it in radians or degrees? Between which values of theta is the formula valid?

L122: Please indicate delta in Fig. 5.

L125/126: Please, compare the equivalent section radius of a single ring with the one of Grassl (2002).

Section 2 and regarding my previous comment to your original Page6: You answered that you did not take into account rebound. This is ok. But my comment had different meaning not looking at the overall rebound of block in the net. If you take a single ring that previously has been plastically deformed and you cut it in one place, then you can observe an inward snapping of the cut/open ends (see figure 3.6 of https://www.research-collection.ethz.ch/bitstream/handle/20.500.11850/148332/eth-27491-02.pdf). This shows that a certain amount of elastic energy has been stored within the deformed ring. Please, take this amount of energy into your energy balance to adequately solve the mechanics.

Fig.11:

- If you remove the boundary conditions parallel to the edges (you can leave a single one for numerical stability) than you get exactly the boundary conditions as you would have in the field with the net supported along a rope.

- Remove subfigre 11(a)

Figs. 12 & 13: "Both" –> "Two"

L247: "by 1m/s, the impact velocity of rockfall is v_lim+1, at this point," –> "to v_lim + 1m/s" L358: DANY –> DYNA (also stated in my previous comments)

Table 3: Please discuss and compare whether - and if so how - static sag of the net has been considered in each case.

Section 2.4: Add the comparison with Grassl (2002) to your discussion!

---

## Referee Comment (RC3) · Anonymous Referee #2 · 13 Sep 2018

General comments The manuscript does not make a significant contribution to the understanding of a natural hazard. It mainly deals with one element of a rockfall barrier. In the manuscript a single ring of a net with its deformation properties is analyzed.

In the abstract it is promised that this ring will be exposed to a two-point, a four-point and a six-point stress and these results will be confirmed by experiments. Such results for all three loads cannot be found in the manuscript.

The introduction should be better structured and should be focused on the title of the manuscript. It does not make sense to describe the simulations of complex rockfall barrier if in the title is mentioned "ring net" and in the manuscript only one single ring is analysed under special conditions.

[Figure]

Specific comments In the theoretical part, the single ring is treated as one element. In reality, however, a ring can be made of different materials (wire bundles, ropes, etc.). The properties of the material used are decisive for the deformation properties of a ring. The manuscript does not address these important differences and therefore is of little interest to design engineers.

A small rockfall barrier may have around 500 individual rings which are hung together. Between the individual rings occur forces and deformations, which must be considered in the simulation of ringnets. These constraints and their influence on the results are not dealt with. In the manuscript two types of arrangement of rings in protective nets are drawn. This arrangement is crucial in the deformation of ringnets. The diagonal arrangement produces much more deformation than the orthogonal arrangement. This problem is likewise not addressed by the authors.

In a subchapter the influence of the inclination angle of the stone trajectory on the net is simulated. But not the trajectory of the stone is changed, but the inclination of the net. This is fundamentally wrong, since the gravitational force has a smaller influence on the deformation (load) of the ringnet in a flatter trajectory. Furthermore, it is shown in a simulation that forces are to be orthogonally removed in diagonally arranged rings in the network Twenty years of testing ringnets in Switzerland, France and Italy show the opposite result. Obviously there is fundamental error in the simulation results.

Technical corrections The introduction is a collection of work done with ring nets and complete rockfall barrier. The simulations of complete protection nets should be omitted and describe the work with ring nets better and more detailed. Line 40: In Figure 3 (a) a ringnet is shown with orthogonally arranged rings and in figure 4 (a) one with diagonal aranged rings. The different behavior should be explained in the manuscript. Line 43: In Spadari et al. (2012) no ring nets are treated but wire mesh. Therefore, this reference should be omitted. Line 46: Thoeni et al. (2013) Also in this publication wire meshs are simulated and therefore this reference does not contribute to a better understanding of ring nets. Omit. Line 99: The terms EF and CD are to be explained

in the text or otherwise omitted. Line 106: Why are "delta"and "phi" not drawn in Fig. 5? Please perform. Line 109: In the whole manuscript must be more clearly distinguished, where a single wire (in the manuscript translated as "coils") and where a ring with many wires is meant. The translation of a single wire with "coil" is wrong. Line 124: The remarks added above should apply mutatis mutandis to Chapter 2.2. Line 147: Also in chapter 2.3 the values "delta" and "phi" are not shown. Line 171: The results presented in chapter 2.4 can't be verified because the mentioned China internal publications are not publicly available. Line 180: With the simple principles presented in the previous chapters, the simulation (Chapter 3) of the experiments (Grassl et al., 2002) can't be explained. In a simulation must be specified with what forces the rings are attached. Depending on the height of the forces occur larger or smaller deformations of the ringnet. Also, the friction between the rings has an influence on the deformation. These points are not discussed. The results of the simulation (displacement, acceleration, impact time) will be presented and the simulation itself will not be described. Above all, in fig.6 the single ring is loaded orthogonally and diagonally in the simulation. Line 208: The individual rings are orthogonally interconnected in Fig. 11, although the ring R7 / 3/300 has never been so arranged. The rings should be drawn as in Fig. 10. It should also be indicated how many rings are actually installed. Line 213: In the simulation with the three boundary conditions, indicate how much the rope is tensioned. In addition, I find it an impertinence for the reader that he should read the results himself from Figures 12 and 13. It is the task of the author to show and explain his results in detail. Here I expect next to the diagrams also a table with the results about the deformation and the braking time of the three simulations. Line 220: A mass of 830 kg at a velocity of 7 m / s produces a kinetic energy of 20.3 kJ and not a value between 16-20 kJ as shown in Figure 13. Here in Fig. 13, there must be an obvious error. In addition, the position of the mass must be taken into account in an energy analysis. It would be better to represent the energy of the mass with respect to the greatest deformation. In addition to the representation of the deformation, the speed and the deceleration could also be displayed. These additional diagrams would

make the simulation more transparent. Line 231: The specification of the maximum energy with two decimal places are exaggerated. Here are integer values. Line 239: If the influence of the angle of incidence $\alpha$ is to be investigated in this chapter, then the trajectory of the mass should also be changed in this sense and not the ring net as shown in Fig. 16 should be rotated. The results will certainly be different, because the influence of gravitational acceleration decreases with increasing angle a. Line 265: The theoretical basics and simulations described in this chapter are extrapolated from the single ring to a net with 6 * 12 rings (Fig.17). At the edge of the rings are attached to ropes and in the ropes energy is reduced. This aspect is not mentioned. It is only claimed that the difference of the energy of the numerical simulation and the theoretical calculation is only 0.71%. Line 290: Why are not both energies (bending and tensile) added together in the two stripes? Please explain. Line 321: The correctness of the theoretical results is not confirmed for rings with 6 adjacent rings. Line 323: The presented numerical model can't be used to model a complete protection net with supports and ropes, but only a part of a ring net which is stored under certain conditions. Line 326: It is not the trajectory inclination that has been changed, but the position of the (horizontal) ring net. Therefore, the results should be interpreted with caution. Line 330: Passive protective nets against falling rocks as shown in Fig. 2 and Fig.3 consist of substantially more elements than described in this manuscript. The most important element of modern safety nets are the brake elements installed in the ropes to limit the forces.

In conclusion: No protective net can be calculated with the methods presented here, not even theoretically. I recommend that the manuscript be withdrawn, completely re-edited and resubmitted.

Please also note the supplement to this comment:
https://www.nat-hazards-earth-syst-sci-discuss.net/nhess-2018-76/nhess-2018-76-RC3-supplement.pdf

---

## Author Comment (AC2) · 20 Sep 2018

Dear Referee, Thank you for your comments concerning our manuscript entitled "Study on Mechanical Properties and Dissipation Capacity of Ring Net in Passive Rockfall Barriers" (Manuscript Number: nhess-2018-76). Those comments are all valuable and very helpful for revising and improving our paper, as well as the important guiding significance to our researches. We have studied comments carefully and have made correction. We hope these revisions will meet with approval. The main corrections in the paper and the responds to your comments are as follows:

L44ff: Your judgement on the existing research is to general. - There is still more existing research to include in this overview section. Please, intensify your literature

[Figure]

research. It is not my part to list all the single research works that exist since the late nineties. For example, the original works of Nicot were not published in 2012 but much earlier in the late nineties and at the beginning of the millenium. Or Volkwein (2004) setup a special discrete element for net rings ("Volkwein, A. (2004). Numerische simulation von flexiblen steinschlagschutzsystemen (No. 289).vdf Hochschulverlag AG."). And much more publications exist..... Response: Thank you for your good advice. the original works of Nicot we quoted was published in 2001. Volkwein (2004) setup a special discrete element for net rings, The specially developed software application Faro simulates the dynamic behaviour of a spherical rock stopped by such a protection barrier in many short time-steps by the central differences method. This enables a detailed view of the dynamics of the modelled barrier and also provides information on its loading and degree of utilisation. The results of the simulations are compared to the field tests carried out within the research project. We have added their research to the paper.

- You wrote that there is barely no difference in the existing research. Please, explain both what is the same between these researches and what is the difference of your research to the existing ones. From my point of view, your research is pretty much the same. Response: Most of the current research focuses on field tests and numerical simulations, in this paper, we analyzed the mechanical properties and energy dissipation of the ring network in the overall protective structure. Firstly, the energy dissipation formula of a single ring is calculated, and then extended to the entire ring network. It is expected to provide certain theoretical basis and guidance for the design of the overall protective structure.

L46: After reading this sentecne I would expect that your article finally brings the adequate mechanics. However, it still lacks a lot (three point tension, comparison with the analytical solution of Nicot (1998/99) etc.). Response: Thank you for your good advice. The problem of three point tension is that we need to study further next.However, this paper mainly studies the mechanical properties and dissipation capacity of the ring net.

The proportion of the 3-point connecting ring is relatively small, which has little effect on the energy consumption of the whole net.

L75: "Grassl hans gerhard" –> "Grassl" Response: Thank you for your good advice. We have revised it in the paper.

L108: Insert "Point" between "Two Tension" (same as in L140). Response: Thank you for your good advice. We have revised it in the paper.

Section2: In your reply to my previous comments regarding 3-point-tensioned rings you stated that the influence of these rings is marginal. However, your calculation examples contain 20-33% of rings that are connected at three and not four points. I would estimated that this number is not small. Please quantify the influence and error induced by this assumption. Response: Thank you for your good advice. However, in the actual calculation process, we mainly calculated the energy consumption of the cross region, and the 3-point-tensioned rings only accounted for a small part of the area. Therefore, we thought the influence of these rings was marginal.

Fig.5: Where are the points CDEF? Are they need anyhow? Do you need ER & CD in this figure? Response: Thank you for your good advice. We have revised it in the paper, the point F has been removed.

L122: Which unit has to be taken for theta? Is it in radians or degrees? Between which values of theta is the formula valid? Response: Thank you for your good advice. The unit of theta should be taken for degrees, at this point, the value of theta is the formula valid.

L122: Please indicate delta in Fig. 5. Response: Thank you for your good advice. $\delta$=BF, We have revised it in the paper.

L125/126: Please, compare the equivalent section radius of a single ring with the one of Grassl (2002). Response: Thank you for your good advice. this paper take the R7 / 3/300 ring as an example, a single ring with a diameter of 300mm is formed by a 3mm

steel wire wound around 7 laps. In the actual production process, the single ring may be wrapped unevenly, so we adopt the concept of equivalent section radius, which is also used in the numerical simulation. However, the concept of equivalent radius is not used in Grassl (2002).

Section 2 and regarding my previous comment to your original Page6: You ansered that you did not take into account rebound. This is ok. But my comment had different mean ing not looking at the overall rebound of block in the net. If you take a single ring that previously has been plastically deformed and you cut it in one place, then you can observe an inward snapping of the cut/open ends (see figure 3.6 of https://www.researchcollection.ethz.ch/bitstream/handle/20.500.11850/148332/eth-27491-02.pdf). This shows that a certain amount of elastic energy has been stored within the deformed ring. Please, take this amount of energy into your energy balance to adequately solve the mechanics. Response: Thank you for your good advice. What you said about the elastic energy stored in the deformation ring is the problem we need to study next, which is our next research direction.

Fig.11: - If you remove the boundary conditions parallel to the edges (you can leave a single one for numerical stability) than you get exactly the boundary conditions as you would have in the field with the net supported along a rope. - Remove subfigre 11(a) Response: Thank you for your good advice. In this paper, we simplify the boundary condition and simplify them into three forms: four-sided fixation, two-sided fixation and four-corners fixation.

Figs. 12 & 13: "Both" –> "Two" Response: Thank you for your good advice. We have revised it in the paper.

L247: "by 1m/s, the impact velocity of rockfall is v_lim+1, at this point," –> "to v_lim +1m/s" L358: DANY –> DYNA (also stated in my previous comments) Response: Thank you for your good advice. We have revised it in the paper.

Table 3: Please discuss and compare whether - and if so how - static sag of the net

has been considered in each case. Response: Thank you for your good advice. The problem of static sag of the net is the next problem we need to study, which is our next research direction.

Section 2.4: Add the comparison with Grassl (2002) to your discussion! Response: According to the results of Grassl (2002), the energy consumption of a single ring under four-point tension is about 10kJ, which is quite different from the results we calculated. However, according to the results of wang (Wang, M.: Rockfall impact protection system, Ph. D. Diss, Chongqing, Logistical Engineering University, 2011.), the energy consumption is 0.89kJ. That's pretty close to our calculation, this is an interesting question and we need to study it further.

Please also note the supplement to this comment:
https://www.nat-hazards-earth-syst-sci-discuss.net/nhess-2018-76/nhess-2018-76-AC2-supplement.zip

---

## Author Comment (AC3) · 20 Sep 2018

Dear Referee, Thank you for your comments concerning our manuscript entitled "Study on Mechanical Properties and Dissipation Capacity of Ring Net in Passive Rockfall Barriers" (Manuscript Number: nhess-2018-76). The main corrections in the paper and the responds to your comments are as follows:

The manuscript does not make a significant contribution to the understanding of a natural hazard. It mainly deals with one element of a rockfall barrier. In the manuscript a single ring of a net with its deformation properties is analyzed. Response: The passive flexible protection system is mainly used to prevent falling rock disasters and protect people's life and propertyïijŇin this paper, we analyzed the mechanical properties and

energy dissipation of the ring network in the overall protective structure. Firstly, the energy dissipation formula of a single ring is calculated, and then extended to the entire ring network. It is expected to provide certain theoretical basis and guidance for the design of the overall protective structure.

In the abstract it is promised that this ring will be exposed to a two‐point, a four‐point and a six point stress and these results will be confirmed by experiments. Such results for all three loads cannot be found in the manuscript. Response: The formula of the single ring under the two‐point, a four‐point and a six- point stress was pushed forward and compared with the experimental data in the literature to verify the correctness of the formula.

The introduction should be better structured and should be focused on the title of the manuscript. It does not make sense to describe the simulations of complex rockfall barrier if in the title is mentioned "ring net" and in the manuscript only one single ring is analysed under special conditions. Response: Numerical simulation is mentioned in the introduction to illustrate the research status, and we first calculated the energy dissipation formula of a single ring and then extended to the entire ring network.

In the theoretical part, the single ring is treated as one element. In reality, however, a ring can be made of different materials (wire bundles, ropes, etc.). The properties of the material used are decisive for the deformation properties of a ring. The manuscript does not address these important differences and therefore is of little interest to design engineers. Response: In actual engineering, most of the materials of the ring net are steel, and the ring we studied is mainly R7/3/300.

A small rockfall barrier may have around 500 individual rings which are hung together. Between the individual rings occur forces and deformations, which must be considered in the simulation of ring nets. These constraints and their influence on the results are not dealt with. In the manuscript two types of arrangement of rings in protective nets are drawn. This arrangement is crucial in the deformation of ring nets. The diagonal

arrangement produces much more deformation than the orthogonal arrangement. This problem is likewise not addressed by the authors. Response: In this paper, we have studied the constraint conditions. The arrangement of the two types is the problem we need to further study, which is our next research direction.

In a subchapter the influence of the inclination angle of the stone trajectory on the net is simulated. But not the trajectory of the stone is changed, but the inclination of the net. This is fundamentally wrong, since the gravitational force has a smaller influence on the deformation (load) of the ring net in a flatter trajectory. Furthermore, it is shown in a simulation that forces are to be orthogonally removed in diagonally arranged rings in the network Twenty years of testing ringnets in Switzerland, France and Italy show the opposite result. Obviously there is fundamental error in the simulation results. Response: We use the inclination of the ring net to represent the change of the falling rock trajectory. Different falling rock trajectories have different consumption of falling rock energy, with the increase of the impact angle, the destruction method of the ring net is the fracture of the central ring at the connection of the lower support rope.

The introduction is a collection of work done with ring nets and complete rockfall barrier. The simulations of complete protection nets should be omitted and describe the work with ring nets better and more detailed. Response: The simulations of complete protection nets is mentioned in the introduction to illustrate the research status, and we mainly write about the work of the ring nets.

Line 40: In Figure 3 (a) a ring net is shown with orthogonally arranged rings and in figure 4 (a) one with diagonal arranged rings. The different behavior should be explained in the manuscript. Response: Thank you for your good advice. Although there are two types of arrangements, the energy dissipation performance of a single ring is not affected.

Line 43: In Spadari et al. (2012) no ring nets are treated but wire mesh. Therefore, this reference should be omitted. Response: Thank you for your good advice. We have

omitted it.

Line 46: Thoeni et al. (2013) Also in this publication wire meshs are simulated and therefore this reference does not contribute to a better understanding of ring nets. Omit. Response: Thank you for your good advice. We have omitted it.

Line 99: The terms EF and CD are to be explained in the text or otherwise omitted. Response: Thank you for your good advice. We have revised it in the paper, the point F has been removed.

Line 106: Why are $\delta$ and $\theta$ not drawn in Fig. 5? Please perform. Response: Thank you for your good advice. $\delta$=BF, we have revised it in the paper.

Line 109: In the whole manuscript must be more clearly distinguished, where a single wire (in the manuscript translated as "coils") and where a ring with many wires is meant. The translation of a single wire with "coil" is wrong. Response: Thank you for your good advice. We have revised it in the paper.

Line 124: The remarks added above should apply mutatis mutandis to Chapter 2.2. Response: Thank you for your good advice. I describe it in Chapter 2.1, and we use the same method for energy calculation in Chapter 2.2.

Line 147: Also in chapter 2.3 the values $\delta$ and $\theta$ are not shown. Response: Thank you for your good advice. $\theta$ has been shown in chapter 2.3, $\delta$ is the radial displacement variation under tension load on diameters.

Line 171: The results presented in chapter 2.4 can't be verified because the mentioned China internal publications are not publicly available. Response: Thank you for your good advice. However, we did not see relevant experimental data in other literatures.

Line 180: With the simple principles presented in the previous chapters, the simulation (Chapter 3) of the experiments (Grassl et al., 2002) can't be explained. In a simulation must be specified with what forces the rings are attached. Depending on the height of the forces occur larger or smaller deformations of the ring net. Also, the friction

between the rings has an influence on the deformation. These points are not discussed. The results of the simulation (displacement, acceleration, impact time) will be presented and the simulation itself will not be described. Above all, in fig.6 the single ring is loaded orthogonally and diagonally in the simulation. Response: Thank you for your good advice. We simplified the model during the simulation and assumed it was in an ideal state. he arrangement of the two types is the problem we need to further study, which is our next research direction.

Line 208: The individual rings are orthogonally interconnected in Fig. 11, although the ring R7/ 3/300 has never been so arranged. The rings should be drawn as in Fig. 10. It should also be indicated how many rings are actually installed. Response: Thank you for your good advice. This section mainly studies the influence of boundary conditions on the passive protective net, and the number of rings is useless.

Line 213: In the simulation with the three boundary conditions, indicate how much the rope is tensioned. In addition, I find it an impertinence for the reader that he should read the results himself from Figures 12 and 13. It is the task of the author to show and explain his results in detail. Here I expect next to the diagrams also a table with the results about the deformation and the braking time of the three simulations. Response: Thank you for your good advice. We have revised it in the paper.

Line 220: A mass of 830 kg at a velocity of 7 m / s produces a kinetic energy of 20.3 kJ and not a value between 16‐20 kJ as shown in Figure 13. Here in Fig. 13, there must be an obvious error. In addition, the position of the mass must be taken into account in an energy analysis. It would be better to represent the energy of the mass with respect to the greatest deformation. In addition to the representation of the deformation, the speed and the deceleration could also be displayed. These additional diagrams would make the simulation more transparent. Response: Thank you for your good advice. We simplified the model during the simulation and assumed it was in an ideal state. Therefore, the numerical simulation energy will be less.

Line 231: The specification of the maximum energy with two decimal places are exaggerated. Here are integer values. Response: Thank you for your good advice. We have revised it in the paper.

Line 239: If the influence of the angle of incidence $\alpha$ is to be investigated in this chapter, then the trajectory of the mass should also be changed in this sense and not the ring net as shown in Fig. 16 should be rotated. The results will certainly be different, because the influence of gravitational acceleration decreases with increasing angle Response: Thank you for your good advice. We use the inclination of the ring net to represent the change of the falling rock trajectory, and the acceleration of gravity is decomposed along the Angle direction

Line 265: The theoretical basics and simulations described in this chapter are extrapolated from the single ring to a net with 6 * 12 rings (Fig.17). At the edge of the rings are attached to ropes and in the ropes energy is reduced. This aspect is not mentioned. It is only claimed that the difference of the energy of the numerical simulation and the theoretical calculation is only 0.71%. Response: Thank you for your good advice. In practical engineering, the ropse need consume energy, but we assume in the simulation that the ropes are completely rigid and does not consume energy, at this time, energy is consumed by the ring net..

Line 290: Why are not both energies (bending and tensile) added together in the two stripes? Please explain. Response: Thank you for your good advice. The length of the transverse strip region is longer, the force on the distant ring is smaller, and it may not enter the plastic stage, while the vertical strip region is more stressed, the ring is easy to enter the plastic stage, so it is assumed that the rings in the vertical strip region all reach plastic deformation, the bending deformation energy of the rings in the transverse strip region, we do the calculations separately, and then we add them up.

Line 321: The correctness of the theoretical results is not confirmed for rings with 6 adjacent rings. Response: Thank you for your good advice. To a certain extent, the

results are acceptable.

Line 323: The presented numerical model can't be used to model a complete protection net with supports and ropes, but only a part of a ring net which is stored under certain conditions. Response: Thank you for your good advice. This manuscript mainly studies the energy consumption of a single ring and a ring nets , the numerical simulation is mainly aimed at the ring net, in order to provide a reference for the design of the passive protective nets.

Line 326: It is not the trajectory inclination that has been changed, but the position of the(horizontal) ring net. Therefore, the results should be interpreted with caution. Response: Thank you for your good advice. We use the inclination of the ring net to represent the change of the falling rock trajectory and the acceleration of gravity is decomposed along the Angle direction, to a certain extent, the results are acceptable.

Line 330: Passive protective nets against falling rocks as shown in Fig. 2 and Fig.3 consist of substantially more elements than described in this manuscript. The most important element of modern safety nets are the brake elements installed in the ropes to limit the forces. Response: Thank you for your good advice. Fig.2 and fig.3 are just for people to have a visual understanding of the passive protective net, and I introduced the composition of the passive protective net in the manuscript.

Please also note the supplement to this comment:
https://www.nat-hazards-earth-syst-sci-discuss.net/nhess-2018-76/nhess-2018-76-AC3-supplement.zip